# Identification and In Silico Characterization of a Conserved Peptide on Influenza Hemagglutinin Protein: A New Potential Antigen for Universal Influenza Vaccine Development

**DOI:** 10.3390/nano13202796

**Published:** 2023-10-20

**Authors:** Atin Khalaj-Hedayati, Seyedehmaryam Moosavi, Otilia Manta, Mohamed H. Helal, Mohamed M. Ibrahim, Zeinhom M. El-Bahy, Ganden Supriyanto

**Affiliations:** 1Department of Chemistry, Faculty of Science and Technology, Airlangga University, Mulyorejo, Surabaya 60115, Indonesia; 2School of Biosciences, Faculty of Health and Medical Sciences, Taylor’s University, Subang Jaya 47500, Malaysia; 3Department of Nanotechnology Engineering, Faculty of Advance Technology and Multidiscipline, Airlangga University, Mulyorejo, Surabaya 60115, Indonesia; m.moosavi1987@gmail.com; 4Romanian Academy, Victor Slavescu Centre for Financial and Monetary Research, 050731 Bucharest, Romania; otilia.manta@rgic.ro; 5Romanian Academy, CE-MONT Mountain Economy Center, 725700 Vatra Dornei, Romania; 6Research Department, Romanian American University, 012101 Bucharest, Romania; 7Department of Chemistry, Faculty of Arts and Science, Northern Border University, Rafha 76413, Saudi Arabia; mohammed.hlal@nbu.edu.sa; 8Department of Chemistry, College of Science, Taif University, P.O. Box 11099, Taif 21944, Saudi Arabia; ibrahim@tu.edu.sa; 9Department of Chemistry, Faculty of Science, Al-Azhar University, Nasr City 11884, Egypt; zeinelbahy@azhar.edu.eg

**Keywords:** epitope mapping, immunoinformatic, hemagglutinin, nanoparticle, peptide-based vaccine, universal influenza vaccine

## Abstract

Antigenic changes in surface proteins of the influenza virus may cause the emergence of new variants that necessitate the reformulation of influenza vaccines every year. Universal influenza vaccine that relies on conserved regions can potentially be effective against all strains regardless of any antigenic changes and as a result, it can bring enormous public health impact and economic benefit worldwide. Here, a conserved peptide (HA2_88–107_) on the stalk domain of hemagglutinin glycoprotein is identified among highly pathogenic influenza viruses. Five top-ranked B-cell and twelve T-cell epitopes were recognized by epitope mapping approaches and the corresponding Human Leukocyte Antigen alleles to T-cell epitopes showed high population coverage (>99%) worldwide. Moreover, molecular docking analysis indicated that *VLMENERTL* and *WTYNAELLV* epitopes have high binding affinity to the antigen-binding groove of the HLA-A*02:01 and HLA-A*68:02 molecules, respectively. Theoretical physicochemical properties of the peptide were assessed to ensure its thermostability and hydrophilicity. The results suggest that the HA2_88–107_ peptide can be a promising antigen for universal influenza vaccine design. However, in vitro and in vivo analyses are needed to support and evaluate the effectiveness of the peptide as an immunogen for vaccine development.

## 1. Introduction

Influenza is considered as of the main human health problems in the world that causes respiratory diseases with 3 to 5 million cases of severe illness and up to 500,000 deaths reported every year [1]. Influenza A virus is classified into subtypes based on two surface antigens, neuraminidase (NA) and hemagglutinin (HA) glycoproteins. Up to now, 18 different HA and 11 different NA proteins have been identified [2]. The pathogenic influenza A virus strains that are currently circulating in the human population include H1N1 and H3N2 and the strains that have a high probability for future pandemic outbreaks are H7N9, H9N2, H5N1, and H2N2 [3,4]. The most effective method to prevent influenza infection is vaccination and currently, most influenza vaccines focus on raising antibodies against the HA glycoprotein. This surface glycoprotein regulates the penetration of the virus into the host cells and it consists of a globular head (HA1) and a stalk region (HA2) [5,6].

Antigenic changes in surface proteins may cause the emergence of new variants of the influenza virus that necessitate the reformulation of influenza vaccines every year [7]. The lack of proof-reading activity of the viral polymerase enzyme is the main reason for changes in the genome, leading to variation in their surface antigens [8]. These antigenic changes allow the influenza viruses to escape host antibody immunity built up through previous vaccination or exposure and decrease the vaccine efficiency [9]. Hence, the circulating subtypes must be recognized annually to reformulate the influenza vaccine. The process relies on surveillance, genotype sequencing, and the measurement of the antigenic properties of the circulating strains. Sometimes the selected strains for vaccine composition may not match with the near future influenza viruses well enough, which may result in increased clinical cases and reduced vaccine protection [9,10]. Moreover, the emergence of new variants with distinct antigenic properties may happen due to the genetic reassortment that can bring up the pandemic with considerable illness and mortality [11]. Currently, the efficacy of the current influenza vaccines is at 44%, therefore, the National Institute of Allergy and Infectious Diseases recently recognized the development of an influenza vaccine with an efficacy of at least 75% as a high scientific priority in which epidemiological impacts of seasonal influenza would be reduced significantly [12]. Hence, the primary goal in influenza vaccine development is to direct the immune system to induce effective responses against multiple subtypes of the virus despite antigenic differences. The strategy to achieve this target is based on modern subunit vaccine development that uses conserved epitopes as vaccine composition to produce a vaccine with broad protection. The main aim is to decrease the occurrence, hospitalization, and death by at least 95%, which would also save $3.5 billion each year in direct medical costs related to influenza. This economic benefit exceeds the proposed and current $330 million in funding for the development of a universal influenza vaccine [13].

Universal vaccine design and development requires precise molecular and physicochemical knowledge of the antigen and its interactions with immune systems. In this regard, in silico tools can provide helpful information to facilitate antigen selection before vaccine development and laboratory experiments [14]. These technologies combine immunogenetics with computational tools to increase antigen selection accuracy based on physicochemical characteristics [15]. Antigen selection, B and T-cell epitope mapping, population coverage, and evaluation of physicochemical properties before conducting in vitro and in vivo experiments, can accelerate the development of the novel influenza vaccine. As studies showed that the antibodies produced against the HA2 domain have a higher neutralization breath compared to HA1 antibodies due to the highest number of conserved epitopes, here, we evaluated HA2 amino acid sequences of six highly pathogenic influenza virus strains (H5N1, H7N9, H9N2, H1N1, H3N2, and H2N2) among human populations by utilizing computational approaches to identify conserved and antigenic peptide [16]. The set of results presented here contributes to the rational design of a universal influenza vaccine based on the conserved peptide that can potentially trigger broad immune responses.

## 2. Materials and Methods

The specific steps involved in identifying and characterizing the influenza conserved peptide are described in detail and a ketch of the entire workflow is demonstrated in Figure 1.

### 2.1. Multiple Sequence Alignment, Structure, and Conservancy Analysis

A total of six highly pathogenic influenza virus strains (H5N1, H7N9, H9N2, H2N2, H1N1, and H3N2) were selected based on the Center for Disease Control and Prevention surveillance and report. The amino acid sequences of the HA2 domain of the selected strains were retrieved from the National Centre for Biotechnology Information (NCBI) database (http://www.ncbi.nlm.nih.gov/protein, access date: 1 July 2023) [3]. The retrieved amino acid sequences were subjected to multiple sequence alignment by Multalin software (http://multalin.toulouse.inra.fr/multalin/, access date: 1 July 2023) to obtain the conserved regions. Interactive 3D structure viewer, iCn3d, (https://www.ncbi.nlm.nih.gov/Structure/icn3d/full.html, access date: 1 July 2023) was used to select a conserved sequence on the HA protein based on the position of the sequence. The conservancy analysis tool from the Immune Epitope Database (IEDB) was used to estimate the identity percentage of the selected peptide in each strain [17]. Also, an NCBI Basic Local Alignment Search Tool (BLAST) analysis (https://blast.ncbi.nlm.nih.gov/Blast.cgi, access date: 1 July 2023) was performed to confirm that the selected sequence is 100% influenza virus-related.

### 2.2. Identification of Linear B-Cell Epitopes

Hydrophilicity, surface accessibility, antigenicity, flexibility, and beta-turn of the candidate peptide were analyzed using different tools from IEDB (http://tools.iedb.org/bcell/, access date: 1 July 2023). The Emini method was used to predict linear B-cell epitope based on surface accessibility [18] and the Karplus and Schulz method was applied for flexibility prediction [19]. Chou and Fasman [20], Parker [21], and Kolaskar and Tongaonkar [22] methods were utilized to predict possible B-cell epitopes based on beta-turn, hydrophilicity, and antigenicity, respectively. Default threshold scores were used for all the methods and the residues with value above the threshold are considered to be part of an epitope.

### 2.3. Identification of T-Cell Epitopes and Population Coverage Analysis

The T-cell epitope prediction tool from IEDB was applied to calculate the binding affinity of the selected peptide to the most frequent Major Histocompatibility Complex (MHC) class I and II in the human population based on software recommendation (http://tools.iedb.org/main/tcell/, access date: 1 July 2023). For the best T-cell epitope selection among all predicted epitopes, the identified epitopes with percentile rank less or equal to 1% for MHC I and less and equal to 10% for MHC II were considered as the potential MHC binder epitopes [23,24]. Later, respective corresponding alleles to the MHC I and II epitopes on the influenza HA2 domain were evaluated for population coverage against the whole world human population using the IEDB population coverage analysis server (http://tools.iedb.org/population/, access date: 1 July 2023). The area option was set to “world” and the calculation option was set to “Class I and II combined”.

### 2.4. Cluster Analysis of the MHC-Restricted Alleles

The MHCcluster 2.0 server (http://www.cbs.dtu.dk/services/MHCcluster/, access date: 5 July 2023) was used to provide pictorial tree-based visualizations and highly instinctive heat map of the functional alliance between the MHC variants that predicted to interact with the selected peptide. The graphical tree and statistic heat map as analysis outputs describe the functional relationship between the MHC alleles from the overlap prediction binding specificity.

### 2.5. Molecular Docking Analysis

The MHC I alleles with the lowest percentile rank for each corresponding epitope (underlined alleles in Table 4) were subjected to docking analysis using the ClusPro 2.0 server (https://cluspro.bu.edu/login.php, access date: 5 July 2023). The 3D structure of MHC I proteins was retrieved from the Research Collaboratory for Structural Bioinformatics Protein Data Bank (PDB) and was refined by Autodock software (http://autodock.scripps.edu, access date: 5 July 2023) to prepare the pre-docking structure. Water molecules were removed and the supporting structure, beta-microglobulin, was retained due to their function of providing stability to the proteins in the host [25]. Also, the FASTA format of MHC I epitopes was converted into PDB format through OpenBabel GUI version 2.3. software, to analyze the interactions with the corresponding MHC molecules (https://openbabel.org/docs/dev/index.html, access date: 5 July 2023). The PDB files of both the receptors and the epitopes were uploaded to the ClusPro 2.0 server to acquire the desirable complexes in terms of free binding energy and better electrostatic interaction. The best models were selected based on the lower global binding energy. The MHC II alleles were not considered for the docking analysis as the structure of the predicted alleles is not available in the PDB database.

### 2.6. Antigenicity, Allergenicity, and Toxicity Assessment

The VaxiJen 2.0 server (http://www.ddgpharmfac.net/vaxijen/, access date: 5 July 2023) was applied to determine the antigenicity of the selected peptide with the threshold value of 0.4 (default) and the virus was selected as the target organism. The AllergenFP online server (http://www.ddg-pharmfac.net/AllergenFP/, access date: 5 July 2023) was used to evaluate the allergenicity of the peptide and the output indicates the peptide is allergen and non-allergen [26,27,28]. Furthermore, ToxinPred (https://webs.iiitd.edu.in/raghava/toxinpred/design.php, access date: 5 July 2023) was used to estimate the toxicity of the peptide [29]. The SVM (Swiss-Prot) + Motif-based method was set for the prediction method. Negative values indicate non-toxic classified results, while positive values indicate the possible toxicity of the analyzed peptide [30].

### 2.7. Assessment of Physicochemical Properties

The ExPAsy ProtParam (https://web.expasy.org/protparam/, access date: 10 July 2023) tool was used to calculate the physicochemical properties of the conserved peptide, which include molecular weight, theoretical isoelectric point (PI), instability index, amino acid composition, chemical formula, atomic composition, estimated half-life, aliphatic index, and grand average of hydropathicity (GRAVY) [31]. A negative GRAVY value indicates that the peptide is non-polar and hydrophilic. Also, a value below 40 for the instability index means the peptide is stable. Moreover, a higher aliphatic index shows higher thermostability. The solubility of the peptide was evaluated using the Pep-Calc protein calculator based on the PI, the number of charged residues, and the peptide length (https://pepcalc.com/protein-calculator.php, access date: 10 July 2023) [32]. The physicochemical properties were used to estimate the stability, thermostability, and solubility of the selected conserved peptide.

## 3. Results

### 3.1. Selection of Conserved Peptide

The HA2 proteome of six highly pathogenic influenza virus strains among human populations was extracted from the NCBI database. The subtypes, NCBI accession number, and amino acid sequences of the HA2 domains are listed in Table 1. The multiple sequence alignment of the HA2 amino acid sequences showed two regions with the highest conservancy based on the conserved amino acid distribution (Figure 2, green and yellow boxes). The first 23 amino acids of the HA2 sequence (Figure 2, green box), which is known as fusion peptide, is embedded in the interspace between monomers of the HA trimer in the native conformation of the virus [33]. The second region with 20 amino acids in length (Figure 2, yellow box), HA2_88–107_: *DVWTYNAELL VLMENERTLD*, is not located at the virus membrane-associated region (Figure 3A). As the embedded region does not meet the peptide selection requirement for the current study, thus, only the HA2_88–107_ sequence was selected for further analysis. The 3D structure of the HA protein and the position of the HA2_88–107_ sequence are shown in Figure 3B. The conservancy analysis of the HA2_88–107_ peptide showed an average of 84.16% identity for the peptide in the homologous protein sets (Table 2) and the NCBI BLAST analysis confirmed that the conserved peptide was 100% originated from the influenza virus. Thus, the HA2_88–107_ sequence was selected for characterization and analysis as a potential antigen candidate.

### 3.2. Linear B-Cell Epitopes of the HA2_88–107_ Peptide

Emini Surface Accessibility [18], Karplus and Schulz Flexibility [19], Chou and Fasman Beta-turn [20], Parker Hydrophilicity propensity [21], and Kolaskar and Tongaonkar Antigenicity analysis [22] methods from the IEDB server were used to identify linear B-cell epitopes on the HA2_88–107_ peptide (Figure 4). The amino acid sequence from 13 to 18 (*MENERT*) on the HA2_88–107_ peptide showed the highest surface accessibility with a score value of 2.745 (Figure 4A). The HA2_88–107_ peptide had a maximum Karplus and Schulz flexibility score value of 1.053 for the region between 13 to 19 amino acid residues (*MENERTL*) (Figure 4B). The Chou and Fasman secondary structure prediction tool identified an amino acid sequence from 1 to 7 (*DVWTYNA*) of the HA2_88–107_ peptide with the highest score value of 1.034 (Figure 4C). Moreover, Parker hydrophilicity analysis indicated that the most hydrophilic region of the HA2_88–107_ peptide is the amino acid sequence from 14 to 20 (*ENERTLD*) with a score value of 4.686 (Figure 4D). Kolaskar and Tongaonkar’s analysis displayed that the sequence from 7 to 13 (*AELLVLM*) of the HA2_88–107_ peptide had the highest score value of 1.125 (Figure 4E). Generally, the higher the score value for the residues from the threshold might interpret that the amino acid sequence is having a higher probability to be part of an epitope (those residues are colored in yellow in Figure 4). In total, 20 linear B-cell epitopes were identified from which 5 of them were considered top-ranked epitopes based on the values and were tabulated in Table 3. The B-cell epitope identification results predicted that the HA2_88–107_ peptide is likely able to interact with B-cells and/or antibodies.

### 3.3. T-Cell Epitopes of the HA2_88–107_ Peptide and Human Population Coverage

The T-cell epitope prediction tool from IEDB analyzed the MHC I and MHC II binding affinity of the HA2_88–107_ peptide with recommended methods. A total of 6 epitopes with a length of 9–11 mer each for MHC I and 6 epitopes with a length of 15 mer each for MHC II were obtained (Table 4). The MHC epitopes of the HA2_88–107_ were selected based on percentile ranks less or equal to 1% for MHC I and less and equal to 10% for MHC II [23]. These 12 epitopes showed binding affinity to a variety of MHC molecules and subsequently their corresponding alleles, such as various Human Leukocyte Antigen (HLA) A and HLA-B for MHC I and HLA-DP, HLA-DQ, and HLA-DR for MHC II molecules. All the corresponding MHC alleles to the T-cell epitopes were used to calculate the global population coverage of MHC I and II alleles for the HA2_88–107_ peptide through the IEDB analysis server (Table 5). The MHC I alleles corresponding to T-cell epitopes cover more than 79.52% of the world population (Figure 5A). The average number of epitope hits/HLA combinations recognized by the world population was 3.12, and the minimum number recognized by 90% of the population (PC90) was 0.49 (Table 5). The MHC II alleles corresponding to the identified T-cell epitopes exhibited a population coverage rate of more than 99.88% (Figure 5B). The average number of epitope hits/HLA combinations recognized by the population was 13.79 and the PC90 was 7.81 (Table 5).

### 3.4. Cluster Analysis of the MHC-Restricted Alleles

The MHCcluster v2.0 online software was used to perform functional clustering of HLA molecules corresponding to the HA2_88–107_ peptide based on correlations between predicted binding affinities [34]. The software produced a clustering of 9 HLA class I and 6 HLA class II, which were recognized to interact with the predicted epitopes and the output was produced through the conventional phylogenetic method based on sequence data available for different HLA alleles. Figure 6 illustrates the function-based clustering of the HLA alleles (heat map) with red zones demonstrating strong correlation and yellow zones indicating weaker interaction of the alleles.

### 3.5. Molecular Docking

The ClusPro software 2.0 was used for molecular docking analysis to investigate the binding pattern of the 6 MHC I predicted epitopes to the top-ranked corresponding HLA molecules (Table 4). Among the 6 epitopes, two epitopes, *VLMENERTL* and *WTYNAELLV*, interacted with the corresponding HLA molecules, HLA-A*02:01 and HLA-A*68:02, respectively, with high affinity at the binding groove of the molecules based on the lowest global energy and attractive Van Der Waals (VDW) in kcal/mol unit (Figure 7). The binding energy scores for MHC I molecules are shown in Table 4.

### 3.6. Antigenicity and Safety of the HA2_88–107_ Peptide

The total antigenicity prediction score for the HA2_88–107_ peptide was 0.2370 via the VaxiJen 2.0 server which is below the threshold value of 0.4 which indicates the peptide has low antigenicity. In addition, the AllergenFP and ToxinPred outputs suggested the non-allergen and non-toxic nature of the peptide for humans, respectively.

### 3.7. Physicochemical Properties of the HA2_88–107_ Peptide

In silico physicochemical analysis revealed that the molecular weight, PI value, and instability index of HA2_88–107_ peptide were 2.42471 kDa, 3.83, and 1.99, respectively. The peptide is acidic and stable at pH 7 based on the PI and instability index values. Moreover, the result showed the peptide is 45% hydrophobic, 25% acidic, 5% basic, and 25% neutral. The peptide consists of 5 negatively charged residues and 1 positively charged residue. The chemical formula is C_107_H_166_N_26_O_36_S_1_, and the total number of atoms is 336. The half-life was estimated to be 1.1 h in mammalian reticulocytes, 3 min in yeast, and >10 h in *Escherichia coli*. The aliphatic index was estimated to be 112, indicating thermostability. The GRAVY was predicted to be −0.265. A negative GRAVY value indicates that the peptide is non-polar and hydrophilic. The peptide was evaluated as a soluble protein in water based on the PI, the number of charged residues, and the peptide length. However, the only way to determine the exact solubility of a peptide is by laboratory experiments.

## 4. Discussion

The timely development of new vaccines is a scientific challenge to combat the ever-increasing global burden of viral diseases. In the case of the influenza virus, frequent antigenic changes in surface proteins, necessitate the reformulation of influenza vaccines every year [7]. So, the generation of a universal vaccine candidate that works against different pathogenic influenza virus strains could have significant public health and economic effects worldwide as it would surmount the current shortcomings of the annual vaccine and also could provide a cost-effective alternative to the annual influenza vaccine [35]. To achieve this goal, a conserved antigenic peptide needs to be identified. Thanks to advances in sequencing technology, we have abundant information about the genomics and proteomics of influenza virus. Therefore, it is possible to design peptide vaccines based on a conserved epitope using various immunoinformatic and cheminformatic tools [36,37].

In the past century, only H1N1, H2N2, and H3N2 influenza subtypes have caused pandemics, making these HA proteins the priority for vaccine development against future pandemics [38]. The second priority targets for vaccine development are defined by HA proteins from viruses that are known to have sporadically infected humans in the past such as H5N1, H7N9, and H9N2. Although second-group viruses are not easily transmitted to humans, they can cause very high mortality (30–60%) in the human population that is infected with these viruses [39,40]. In the present study, HA2 amino acid sequences of the mentioned six highly pathogenic influenza A viruses among the human population were retrieved from the NCBI database, and conserved regions were identified by multiple sequencing alignments. The HA2_88–107_ peptide was selected from two identified sequences as it was exposed on the surface of the virus compared to the other conserved regions based on 3D structure analysis. Also, it has been shown that the region between amino acids 35 and 107 of the HA2 sequence was accessible to the antibodies and immune cells in the native structure of the virus [41]. In general, antigens that are displayed on the surface of the virus are more likely accessible to the immune system at the first point of the infection [42]. Moreover, the HA2_88–107_ peptide was considered a highly conserved peptide as its average similarity (84.16%) with the homologous protein sets was more than 70%, showing that it may serve as a candidate antigen for the universal influenza vaccine design [43]. Although the H3N2 subtype showed 65% similarity to the HA2_88–107_ peptide (Table 2), the amino acid differences were conservative replacements, which means it is an exchange between two amino acids separated by a small physicochemical distance. It is necessary to check the similarity between the selected peptide and self-proteins while designing vaccine candidates to avoid any cross-reactivity of the induced immune responses [44]. The NCBI BLAST analysis confirmed that the peptide did not have any similarity to self-proteins or antigens from other pathogens and it 100% belongs to the influenza virus.

Epitope mapping has been performed to identify potential B and T-cell epitopes on the conserved HA2_88–107_ peptide to validate the antigen selection for vaccine design. An epitope refers to a specific site on an antigen to which a complementary antibody and/or immune cells may specifically bind. The B-cells, which are part of the adaptive immune system, act as receptors, bind to the antigen’s epitopes, mediate humoral immunity, and have crucial roles in influenza viral infections and vaccinations [45,46]. In general, B-cell epitopes are categorized into linear (continuous) epitopes, which consist of a linear sequence of residues; and conformational (discontinuous) epitopes, which consist of residues that are brought together by folded protein structure and are not continuous in the primary protein sequence [47]. The prediction of linear epitopes has received major attention as they can readily be used as antigens for vaccine and antibody production. In the case of conformational epitopes, 3D structure information and suitable scaffolds for epitope grafting are required. In the current study, only linear epitopes were predicted based on amino acid propensity scales depicting the physicochemical features of B-cell epitopes [48]. Twenty linear B-cell epitopes were identified that indicated the HA2_88–107_ peptide is likely able to interact with B-cells. Among them, five epitopes are considered high-ranked epitopes based on surface accessibility, flexibility, secondary structure, hydrophilicity, and physicochemical properties.

The amino acid sequence from 13 to 18 (*MENERT*) on the HA2_88–107_ peptide showed the highest surface accessibility with a score value of 2.745. The higher surface accessibility of the epitope means a higher chance of being recognized by B-cells or antibodies [18]. The HA2_88–107_ peptide had a maximum flexibility score value of 1.053 for the region from 13 to 19 amino acid residues (*MENERTL*). When the epitope is flexible, several sets of conformations within the native antigen can cause the generation of different antibodies that lead to broader antibody-mediated responses [49]. The Chou and Fasman secondary structure analysis identified an amino acid sequence from 1 to 7 (*DVWTYNA*) with the highest score value of 1.034. The tool predicts the epitope based on the beta-turn structure scale, where the polypeptide chain folds back on itself by nearly 180 degrees and gives a protein its globularity rather than linearity structure and the beta turns in a protein are more surface accessible [20]. In general, the conformation of epitope allows recognition of the epitope by B-cells or antibodies, and in this case, it is thought to be an important aspect of vaccine design [50,51]. Furthermore, the Parker hydrophilicity analysis indicated that the most hydrophilic region of the HA2_88–107_ peptide is the amino acid sequence from 14 to 20 (*ENERTLD*) with a score value of 4.686. As hydrophilic regions are predominantly located on the protein surface, subsequently, they can be considered as more antigenic regions [21,48]. The Kolaskar and Tongaonkar analysis displayed that the sequence from 7 to 13 (*AELLVLM*) of the HA2_88–107_ peptide had the highest score value of 1.125. The method uses the physicochemical properties of amino acids and their frequencies of occurrence in experimentally known segmental epitopes to predict antigenicity [22].

On the other hand, cell-mediated immunity is another part of the adaptive immune response responsible for long-lasting immunity. It can restrict the spread of the infection directly and activate other immune cells to destroy pathogens, respectively [52,53]. Cytotoxic T-cells (CD8+) and helper T-cells (CD4+) play important roles in cell-mediated immunity. They scan other cells through their T-cell receptors for MHC:peptide complexes and recognize epitopes that are presented by MHC molecules [54]. When stimulated with a MHC molecule that presents antigenic peptides, T-cells can directly restrict the spread of infection in human cells and they have been able to provide cross-reactivity in the recognition of the different subtypes of influenza A virus when evoked by conserved regions of the influenza virus [55,56,57]. The epitope hits/MHC I molecule complex can trigger the CD8+ T cells and the epitope hits/MHC II molecule complex can be detected by CD4+ T cells, which is very important in the regulation of both CD8+ T and B-cells [58]. Thus, prediction of interaction between T-cell epitopes and MHC molecules is required in the design of vaccines against pathogens. Here, MHC binding prediction results demonstrated 12 T-cell epitopes on the HA2_88–107_ peptide that can form MHC:epitope complexes to interact with CD8+ T and CD4+ T-cells and are considered potential epitope hits. These epitopes can interact with various MHC molecules in humans such as HLA-A and HLA-B for MHC class I and HLA-DP, HLA-DQ, and HLA-DR for MHC class II molecules. The epitopes were selected based on the percentile rank, which is a transformation that normalizes the binding affinity scores across different MHC molecules and enables MHC binding prediction and comparisons. A lower percentile rank value indicates higher affinity [23,59].

The MHC molecules are highly polymorphic and the expression frequency of different MHC molecules varies in different ethnicities and geographic areas around the world [60,61,62]. This polymorphism is basically in the gene region encoding the binding groove of MHC molecules and causes widely varying binding specificities [63]. Thus, any selected epitope for epitope-based vaccine design should bind to several molecules of HLA supertype for maximum population coverage, especially in the case of universal vaccine development [43]. In general, population coverage is defined as the fraction of individuals in a population that responds to the predicted epitopes of an antigen based on the HLA allele frequencies of the population [64]. In the current study, all the corresponding alleles to the epitopes from T-cell epitope identification analysis were used to indicate the population coverage for the HA2_88–107_ peptide. The results revealed a strong correlation between the corresponding MHC II alleles and the population coverage rate (99.88%). Thereby, the HA2_88–107_ peptide can trigger CD4+ T cells and consequently, regulate CD8+ T and B-cell responses in 99.88% of the world population. Also, based on MHC I predicted alleles, the peptide can trigger CD8+ T cells directly in 79.52% of the world population.

Despite the high polymorphism in the human MHC genomic region, not all MHC molecules are equally different in terms of function. Structure-based clustering methods are effective in identifying superfamilies of MHC proteins with similar binding specificities [65]. As MHC super-families play an important role in drug development and vaccine design, MHC cluster analysis was also applied to specify the functional relationship of MHC variants based on correlations between predicted binding affinities [34]. The MHCcluster v2.0 server, which provides pictorial tree-based visualizations and a heat map of the functional association of MHC variants, generated a cluster of 15 HLA molecules identified as interacting with our predicted epitopes. The output was generated using a standard phylogenetic method based on available sequence data for different HLA alleles. The result of cluster analysis can indicate these alleles are in different cluster groups (more functionality in the population), which can strengthen the prediction from the previous analysis in this study. Furthermore, the molecular docking analysis in vaccine design can help researchers to model the interaction between epitopes and immune cell receptors at the atomic level and facilitate the prediction of the epitope binding possibilities to the immune receptors [66,67]. Hence, a docking study was performed to confirm the association between HLA molecules and our predicted epitopes. Two epitopes, *VLMENERTL* and *WTYNAELLV*, interacted with the corresponding alleles, HLA-A*02:01 and HLA-A*68:02, with high affinity at the binding groove of the HLA molecules. Consequently, the peptide is an ideal vaccine candidate for >1 billion people globally who express HLA-A*02:01 [68].

The VaxiJen 2.0 software classifies antigens based solely on the physicochemical properties of the protein and predicts the antigenicity of an antigen in total, irrespective of epitope mapping. The total antigenicity prediction score for the HA2_88–107_ was below the threshold value of 0.4 which indicates the peptide has low antigenicity [28]. In general, one of the limitations of subunit vaccines is their low antigenicity as they mostly rely on epitopes with the absence of immunostimulatory components from the source pathogens. Different approaches such as the inclusion of nanoparticles and immunopotentiators into vaccine formulation have been applied to enhance the desired antigenicity and immunogenicity of influenza subunit vaccines [69]. For example, the Hepatitis B core Antigen was applied as an immunopotentiator agent to increase the HA2_88–107_ peptide antigenicity [70]. Also, the result from VaxiJen analysis indicated that the total antigenicity score value was increased after the HA2_88–107_ peptide was fused to the N terminal end of the HBcAg protein, which indicates the fusion protein can be antigenic.

The allergenicity assessment is a crucial step in the design of a peptide vaccine as it may induce allergic reactions by initiating immunoglobin E production and enhancing response to histamine [71,72]. The result from AllergenFP software revealed that the HA2_88–107_ peptide was not an allergen to humans so it should not be able to cause any allergenic reaction within the body. The server is a binary classifier between allergens and non-allergens and it predicts allergenicity based on the amino acid principle properties such as size, hydrophobicity, helix, and beta-strand formations [73]. Moreover, high specificity and low cross-reactivity are generally considered to design effective, safe, and theoretically infallible therapeutic molecules and vaccines [30]. In particular, computational screening of non-toxic peptide approaches is needed to improve peptide selectivity with less cost and time [74]. The ToxinPred tool was used to predict and evaluate the toxicity of the HA2_88–107_ peptide in which the negative value indicated that the peptide was not toxic.

Various physicochemical properties of the peptide were analyzed. The instability index was computed below the threshold, which classified the peptide as a stable molecule and ensured that the construct possesses good characteristics to initialize an immunogenic reaction in the body. The value above the threshold indicates the instability of the protein, which can be related to the order of certain amino acids in the sequence [75]. On the other hand, the thermostability of the conserved peptide was demonstrated by the aliphatic index. The index is defined as the relative volume occupied by aliphatic side chains [alanine (Ala), valine (Val), isoleucine (Ile), and leucine (Leu)]. It may be regarded as a positive factor for the increase in thermostability of globular proteins. A high aliphatic index indicates that a protein is thermostable over a wide temperature range [76]. Moreover, the hydrophilic nature of the conserved peptide sequences was represented by the GRAVY value which is calculated as the sum of hydropathy values of all of the amino acids divided by the number of residues in the sequence. A negative GRAVY value indicates that the protein is non-polar and hydrophilic [77]. Furthermore, the PI value and the total number of charges indicated that theoretically, the peptide might dissolve in aqueous media at pH 7. In general, at the PI, proteins have a zero charge and tend to associate together, resulting in insolubility, but at the pH above the PI, the net charge is negative and solubility is likely predictable. On the other hand, the 45% hydrophobicity may cause insolubility or partial solubility of the peptide in an aqueous solution even if the sequence contains more than 25% charged residues [31,77]. So, further in vitro experiment is needed to discover the exact solubility of the conserved peptide.

## 5. Conclusions

In general, the influenza A virus remains a current and future threat to public health and all outbreaks demonstrate the need to develop a universal vaccine to protect against different strains. The computational framework is useful for evaluating the effectiveness of an antigen generated through existing bioinformatics tools. In the current study, the results indicated that the HA2_88–107_ peptide can be a promising antigen for universal influenza vaccine design. However, we recommend further studies based on wet laboratory techniques to validate our predicted peptide experimentally for vaccine design and development.

## Figures and Tables

**Figure 1 nanomaterials-13-02796-f001:**
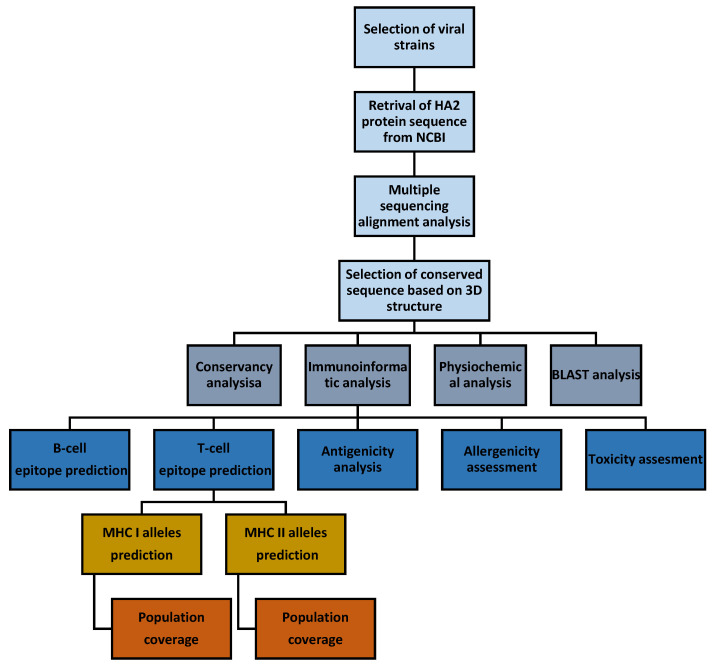
Workflow for in silico prediction and characterization analysis of a conserved peptide on HA glycoprotein as a potential antigen for universal influenza vaccine design.

**Figure 2 nanomaterials-13-02796-f002:**
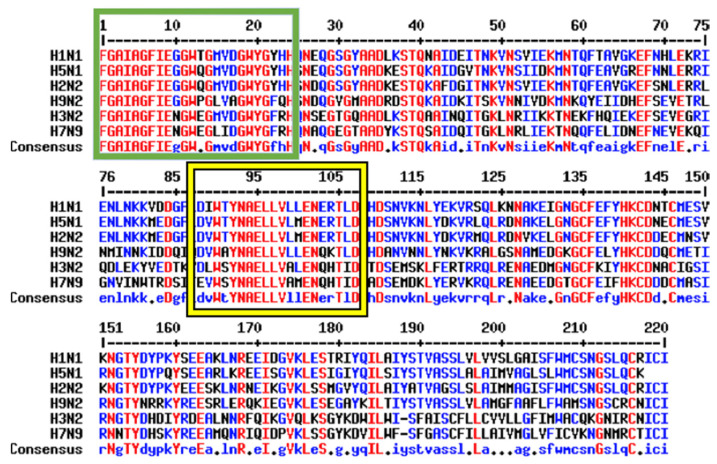
Multiple sequence alignment of HA2 amino acid sequences from six different strains of influenza virus using Multalin online software. The red color demonstrates the full conservancy, the blue color indicates one amino acid alternation and the black color represents the alternation in more than one amino acid at the specific region for different strains. The green (fusion peptide) and yellow (HA2_88–107_) boxes show the regions with the highest conservation distribution. The letter in the consensus line means conservative and the dot indicates non-conservative amino acid substitution.

**Figure 3 nanomaterials-13-02796-f003:**
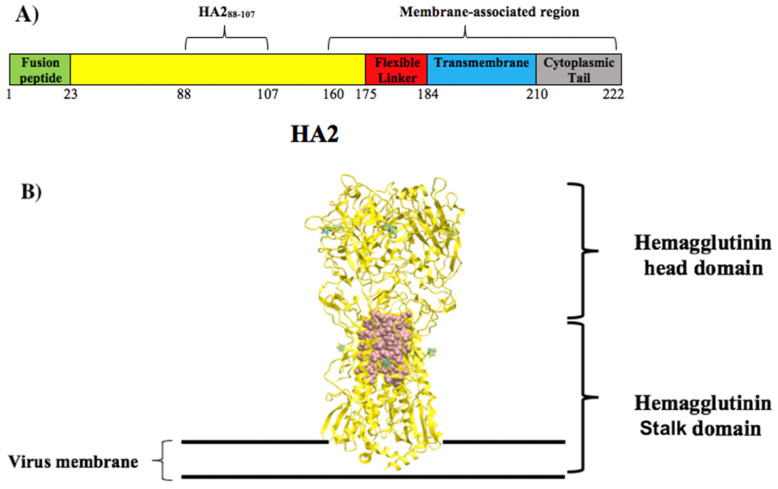
Location of the conserved HA2_88–107_ peptide on influenza HA protein. (**A**) Sequence diagram showing the polypeptide segment of the HA2 domain (222 amino acids) in its primary structure, indicating the HA2_88–107_ peptide is not associated with the membrane. (**B**) The position of HA2_88–107_ peptide (pink color) on the 3D full structure of HA protein (yellow color) using iCn3D online software, access date: 1 July 2023). The green color represents the carbohydrate components of the protein.

**Figure 4 nanomaterials-13-02796-f004:**
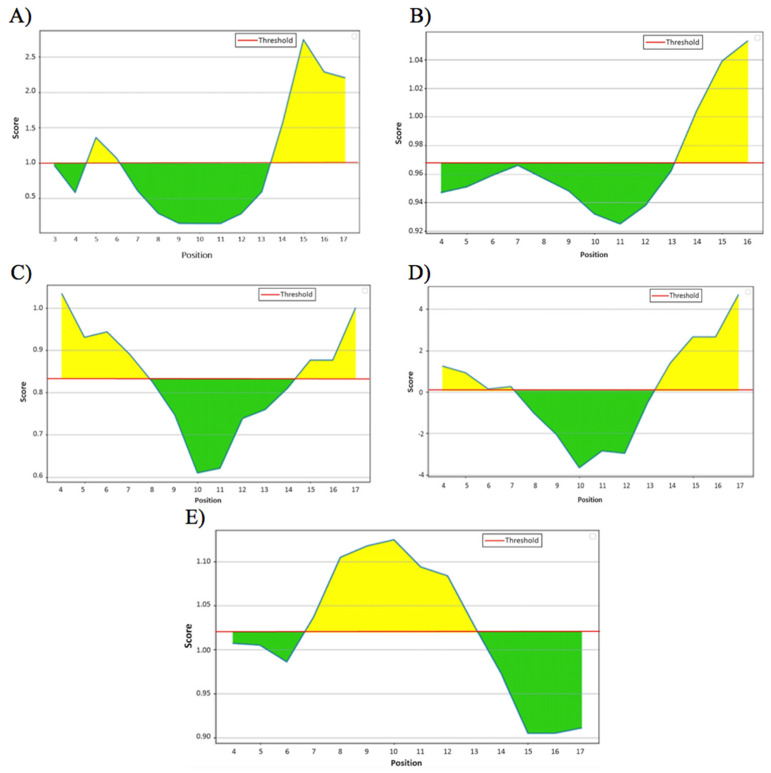
Linear B-cell epitope identification on HA2_88–107_ peptide. The most antigenic epitopes are shown in yellow color above the threshold value (red line) and green areas are not considered as an epitope. (**A**): Emini surface accessibility prediction with a threshold value of 1.000; (**B**): Karplus and Schulz flexibility prediction with a threshold value of 0.968; (**C**): Chou and Fasman secondary structure prediction with a threshold value of 0.834; (**D**): Parker hydrophilicity prediction with the threshold value of 0.064; (**E**): Kolaskar and Tongaonkar antigenicity prediction with the threshold value of 1.020.

**Figure 5 nanomaterials-13-02796-f005:**
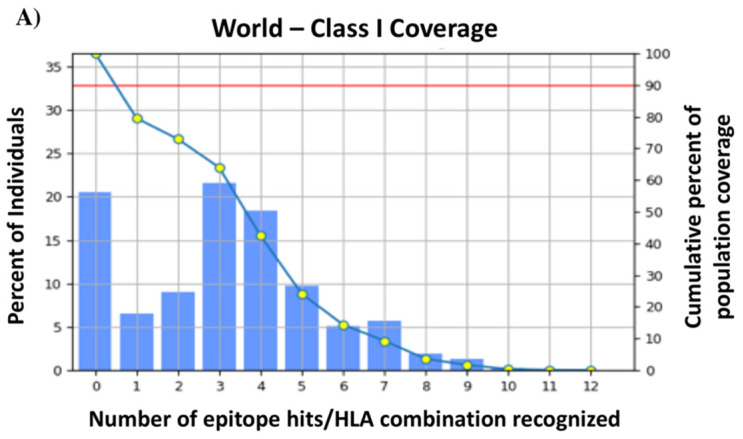
Population coverage of the HA2_88–107_ peptide. The world populations were evaluated for the peptide using IEDB online population coverage analysis. (**A**) The graph shows a population coverage of 79.52% for MHC I epitopes. (**B**) The graph indicates population coverage of 99.88% for MHC II epitopes. The line (-o-) represents the cumulative percentage of population coverage of the epitopes; the bars represent the population coverage for each epitope; the red line represents PC90.

**Figure 6 nanomaterials-13-02796-f006:**
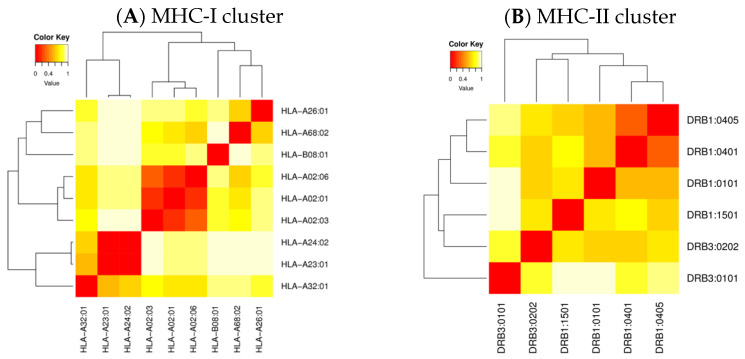
Cluster analysis of the HLA alleles. (**A**) Representing the cluster of MHC-I alleles and (**B**) Representing the cluster of MHC-II alleles. The red color indicates strong interaction and the yellow zone indicates weaker interaction with appropriate annotation.

**Figure 7 nanomaterials-13-02796-f007:**
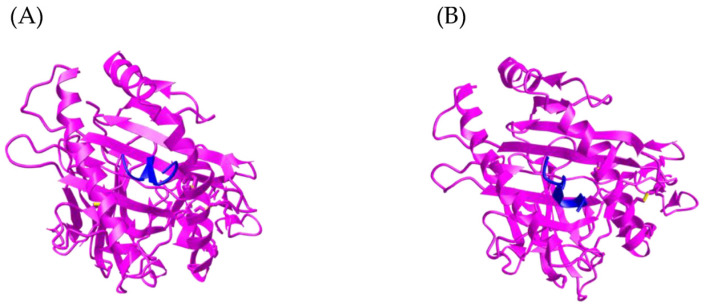
Visualization of ClusPro molecular docking of (**A**) *VLMENERTL* epitope interaction with HLA-A*02:01 molecule and (**B**) *WTYNAELLV* epitope interaction with HLA-A*68:02 molecule. Blue color indicates epitopes and magenta color indicates corresponding molecules (receptors).

**Table 1 nanomaterials-13-02796-t001:** HA2 amino acid sequences of the highly pathogenic influenza A virus with their accession number, geographical regions, and date of collection.

**No.**	Subtype	NCBI Accession Number	FASTA Format of HA2 Amino Acid Sequence
1	H1N1	YP009121767	>YP_009121767.1 FGAIAGFIEGGWTGMVDGWYGYHHQNEQGSGYAADLKSTQNAIDEITNKVNSVIEKMNTQFTAVGKEFNHLEKRIENLNKKVDDGFLDIWTYNAELLVLLENERTLDYHDSNVKNLYEKVRSQLKNNAKEIGNGCFEFYHKCDNTCMESVKNGTYDYPKYSEEAKLNREEIDGVKLESTRIYQILAIYSTVASSLVLVVSLGAISFWMCSNGSLQCRICI
2	H3N2	ASV62273	>ASV62273.1 FGAIAGFIENGWEGMVDGWYGFRHQNSEGTGQAADLKSTQAAINQITGKLNRIIKKTNEKFHQIEKEFSEVEGRIQDLEKYVEDTKVDLWSYNAELLVALENQHTIDLTDSEMSKLFERTRRQLRENAEDMGNGCFKIYHKCDNACIGSIRNGTYDHDIYRDEALNNRFQIKGVQLKSGYKDWILWISFAISCFLLCVVLLGFIMWACQKGNIRCNICI
3	H5N1	ACI06178	>ACI06178.1 FGAIAGFIEGGWQGMVDGWYGYHHSNEQGSGYAADKESTQKAIDGVTNKVNSIIDKMNTQFEAVGREFNNLERRIENLNKKMEDGFLDVWTYNAELLVLMENERTLDFHDSNVKNLYDKVRLQLRDNAKELGNGCFEFYHKCDNECMESVRNGTYDYPQYSEEARLKREEISGVKLESIGIYQILSIYSTVASSLALAIMVAGLSLWMCSNGSLQCK
4	H7N9	AGI60292	>AGI60292.1 FGAIAGFIENGWEGLIDGWYGFRHQNAQGEGTAADYKSTQSAIDQITGKLNRLIEKTNQQFELIDNEFNEVEKQIGNVINWTRDSITEVWSYNAELLVAMENQHTIDLADSEMDKLYERVKRQLRENAEEDGTGCFEIFHKCDDDCMASIRNNTYDHSKYREEAMQNRIQIDPVKLSSGYKDVILWFSFGASCFILLAIVMGLVFICVKNGNMRCTICI
5	H9N2	CAB95857	>CAB95857.1 FGAIAGFIEGGWPGLVAGWYGFQHSNDQGVGMAADRDSTQKAIDKITSKVNNIVDKMNKQYEIIDHEFSEVETRLNMINNKIDDQIQDVWAYNAELLVLLENQKTLDEHDANVNNLYNKVKRALGSNAMEDGKGCFELYHKCDDQCMETIRNGTYNRRKYREESRLERQKIEGVKLESEGAYKILTIYSTVASSLVLAMGFAAFLFWAMSNGSCRCNICI
6	H2N2	AAY28987	>AAY28987.1 FGAIAGFIEGGWQGMVDGWYGYHHSNDQGSGYAADKESTQKAFDGITNKVNSVIEKMNTQFEAVGKEFSNLERRLENLNKKMEDGFLDVWTYNAELLVLMENERTLDFHDSNVKNLYDKVRMQLRDNVKELGNGCFEFYHKCDDECMNSVKNGTYDYPKYEEESKLNRNEIKGVKLSSMGVYQILAIYATVAGSLSLAIMMAGISFWMCSNGSLQCRICI

**Table 2 nanomaterials-13-02796-t002:** HA2_88–107_ peptide identity among the six highly pathogenic strains of the influenza virus.

HA2_88–107_ Sequence	Peptide Identity among the Strains					
	H1N1	H3N2	H5N1	H7N9	H9N2	H2N2
*DVWTYNAELLVLMENERTLD*	90%	65%	100%	70%	80%	100%
	Average: 84.16%					

**Table 3 nanomaterials-13-02796-t003:** Top-ranked linear B-cell epitopes on HA2_88–107_ peptide.

No.	Methods	Epitope Sequence	Start Position	End Position	Score Value	Threshold Value
1	Emini Surface Accessibility	*MENERT*	13	18	2.745	1.000
2	Karplus and Schulz Flexibility	*MENERTL*	13	19	1.053	0.968
3	Chou and Fasman Beta-turn	*DVWTYNA*	1	7	1.034	0.834
4	Parker Hydrophilicity	*ENERTLD*	14	20	4.686	0.064
5	Kolaskar and Tongaonkar Antigenicity	*AELLVLM*	7	13	1.125	1.020

**Table 4 nanomaterials-13-02796-t004:** T-cell epitopes on HA2_88–107_ peptide that bind to the human MHC class I and II alleles with high affinity.

No.	Epitopes	MHC I Allele	Percentile Rank < 1	Binding Energy (kcal/mol)	No.	Epitopes	MHC II Allele	Percentile Rank < 10
1	DVWTYNAEL	HLA-A*68:02 HLA-A*26:01	0.29 0.55	−155.5	1	DVWTYNAELLVLMEN	HLA-DQA1*03:01/DQB1*03:02 HLA-DQA1*04:01/DQB1*04:02 HLA-DQA1*05:01/DQB1*02:01 HLA-DQA1*01:01/DQB1*05:01 HLA-DQA1*01:02/DQB1*06:02 HLA-DPA1*01:03/DPB1*04:01 HLA-DPA1*01:03/DPB1*02:01 HLA-DPA1*02:01/DPB1*01:01 HLA-DPA1*03:01/DPB1*04:02 HLA-DRB3*01:01 HLA-DRB3*02:02	3.7 4.6 4.8 6.3 9.4 3.8 4.4 5.2 7.8 7.4 8.2
2	TYNAELLVL	HLA-A*24:02 HLA-A*23:01	0.08 0.15	−103.2	2	NAELLVLMENERTLD	HLA-DRB1*04:05 HLA-DQA1*03:01/DQB1*03:02 HLA-DRB1*04:01 HLA-DRB1*01:01 HLA-DRB1*15:01	1.9 2.2 3.7 4.9 6.7
3	TYNAELLVLM	HLA-A*24:02 HLA-A*23:01	0.43 0.48	−115	3	TYNAELLVLMENERT	HLA-DQA1*03:01/DQB1*03:02 HLA-DRB1*04:05 HLA-DQA1*04:01/DQB1*04:02 HLA-DRB1*04:01 HLA-DQA1*01:02/DQB1*06:02 HLA-DRB1*15:01	1.5 1.9 3.6 3.7 6.5 8.5
4	VLMENERTL	HLA-A*02:01 HLA-A*02:03 HLA-A*02:06 HLA-A*32:01 HLA-B*08:01	0.1 0.15 0.3 0.46 0.51	−160.8	4	VWTYNAELLVLMENE	HLA-DQA1*03:01/DQB1*03:02 HLA-DQA1*04:01/DQB1*04:02 HLA-DQA1*01:02/DQB1*06:02 HLA-DPA1*01:03/DPB1*02:01 HLA-DQA1*05:01/DQB1*02:01 HLA-DPA1*01:03/DPB1*04:01 HLA-DPA1*02:01/DPB1*01:01 HLA-DPA1*03:01/DPB1*04:02 HLA-DRB3*01:01	1.3 3.1 4.2 4.4 4.4 4.6 5.8 7.4 9.6
5	VWTYNAELL	HLA-A*24:02 HLA-A*23:01	0.54 0.81	−133.8	5	WTYNAELLVLMENER	HLA-DQA1*03:01/DQB1*03:02 HLA-DQA1*04:01/DQB1*04:02 HLA-DQA1*01:02/DQB1*06:02 HLA-DQA1*05:01/DQB1*02:01 HLA-DPA1*02:01/DPB1*01:01 HLA-DPA1*01:03/DPB1*04:01 HLA-DPA1*01:03/DPB1*02:01	1.5 3.5 4.6 4.7 7.9 8.6 10
6	WTYNAELLV	HLA-A*68:02	0.38	−857.9	6	YNAELLVLMENERTL	HLA-DRB1*04:05 HLA-DQA1*03:01/DQB1*03:02 HLA-DRB1*04:01 HLA-DQA1*04:01/DQB1*04:02 HLA-DRB1*01:01 HLA-DRB1*15:01	1.9 2 3.7 5.4 5.4 7.9

**Table 5 nanomaterials-13-02796-t005:** Population coverage of the HA2_88–107_ peptide for MHC I and II alleles.

Population/Area	MHC Class I			MHC Class II		
	Coverage ^a^	Average-Hit ^b^	PC90 ^c^	Coverage ^a^	Average-Hit ^b^	PC90 ^c^
World	79.52%	3.12	0.49	99.88%	13.79	7.81

^a^ projected population coverage. ^b^ average number of epitope hits/HLA combinations recognized by the population. ^c^ minimum number of epitope hits/HLA combinations recognized by 90% of the population.

## Data Availability

The data that support the findings of this study are available on request from the corresponding author.

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
