# Peer review of "Identification and In Silico Characterization of a Conserved Peptide on Influenza Hemagglutinin Protein: A New Potential Antigen for Universal Influenza Vaccine Development"

_nanomaterials, 2023, doi:10.3390/nano13202796_

Round 1
Reviewer 1 Report
Atin Khalaj-Hedayati and colleagues reported an In silico investigation of a conserved peptide on influenza Hemagglutinin protein in order to shed light on potential new antigen for universal vaccine. The authors identified a conserved peptide (HA288-107) on the stalk domain of hemagglutinin glycoprotein among highly pathogenic influenza viruses. By using several epitope mapping approaches 5 B-cell and 12 T-cell epitopes were recognized. Overall, the results suggest that the HA288-107 peptide can be a promising antigen for universal influenza vaccine design. The study is well done according the in silico investigation. However, as stated also by the authors, the experimental proof of the in silico data is lacking. Thus some points should be improved.
Main points
1- lines 300-316: In this paragraph the meaning of population coverage is not clear. This should be explained better also in the materials and methods. Moreover, the percentile ranks less or equal to 1% and 10% used for MHC I and MHC II, respectively, should be explained. The general reader (not expert in this field) have some problems to understand such analysis.
2- The difference of the data obtained in figure 4 E with the other panels A-D should be explained in the text to understand the meaning for the nature of the epitope.
3-Lines 386-387: this paragraph sould be moved after section 3.1. The nature of antigenicity of the peptide should be described before its studies on B-cell and T-cell epitopes.
4- The discussion as it is now is too long. There is too much description of the important of B- and T cell epitopes in the development of a new vaccine that are unnecessary. As stated before, in the absence of experimental data the result is only a weel done in silico study. The fact that the epitope is immunogenic it dosen't mean that have neutralizing antibody epitopes or T-cell citotozxic protective epitopes. Moreover, some paragraph as that in lines 548-562 on the allergenicity is not important at this stage. Collectively, reducing the text to the essential data observed in silico remanding all the consideration to the future experimental data is more appropriate.
Minor points
1- line 112: the subtype of the selected influenza virus taken in consideration (see table 1) should be stated.
2- Figure 3: in the legend it should be stated what mean the green color
3- All typos should be corrected
Reviewer 2 Report
This paper needs some improvement. Indeed, the physicochemical properties of HA288-107 peptide have to be more identified, in particular, I suggest to authors to review these properties by determining:
-The surface tension of the peptide
- The surface charge density
-The isoelectric point and the zero charge point of the peptide
-The Bronsted and Lewis acid-base properties of the peptide.
The effect of the tempertaure on the peptide should be determined, is there any glass transition of this peptide
Round 2
Reviewer 1 Report
The authors have substantially improuved the manuscript and now is approved for publication.
Reviewer 2 Report
The revision of the manuscript is completed and very very well written